# Elastic Stress Field beneath a Sticking Circular Contact under Tangential Load

Emanuel Willert 

Institute of Mechanics, Technische Universität Berlin, Sekretariat C8-4, Straße des 17. Juni 135, 10623 Berlin, Germany; e.willert@tu-berlin.de

**Abstract:** Based on a potential theoretical approach, the subsurface stress field is calculated for an elastic half-space which is subject to normal and uniaxial tangential surface tractions that—in the case of elastic decoupling—correspond to rigid normal and tangential translations of a circular surface domain. The stress fields are obtained explicitly and in closed form as the imaginary parts of compact complex-valued expressions. The stress state in the surface and on the central axis are considered in detail. As, within specific approximations that have been discussed at length in the literature, any tangential contact problem with friction can be understood as a certain incremental series of such rigid translations, the solutions presented here can serve as the basis of very fast superposition algorithms for the analysis of subsurface stress fields in general tangential contact problems with friction. This idea is demonstrated by means of the frictional tangential contact between an elastic half-space and a rigid cylindrical flat punch with rounded corners.

**Keywords:** normal contact; tangential contact; friction; subsurface stress field; potential theory



## 1. Introduction

The solution to a contact mechanical problem is often only concerned with the relations between macroscopic quantities (forces and global displacements), the size and shape of the contact domain, and the traction vector therein. The subsurface stress field beneath the contact is not often considered, at least in exact contact solutions. On the one hand, this is due to the fact that for the analysis of contacts as contributors to the dynamics of a multibody system, the knowledge about the relations between macroscopic quantities is sufficient; on the other hand, the exact analysis of the subsurface stress field is generally quite a complicated mathematical task. A fine example for actually both reasons is the classical paper of Hertz [1]. Based only on his solution for the force and indentation depth in the ("point") contact of smooth elastic bodies, he considered a dynamic multibody problem—the impact of elastic spheres (in the quasi-static limit); on the other hand, the exact solution for the subsurface stresses he considered mathematically close to impossible although undoubtedly acknowledging their importance [2] in various circumstances, e.g., for the analysis of subsurface yield.

Since the work of Hertz, of course, significant (although still quite limited) progress has been made with respect to the exact determination of subsurface stress fields in mechanical contact problems. Theoretically speaking, at least for linearly elastic problems, the knowledge about the traction vector in the contact domain (which commonly is a central part of what is considered a "contact solution") would be sufficient for the determination of the complete stress field based on the superposition of the fundamental solutions (including the corresponding stress fields) by Boussinesq [3] and Cerruti [4] for normal and tangential point loading at the surface of an elastic half-space. However, the resulting integrals are—with very few exceptions—intractable in exact form and computationally costly if evaluated numerically (see, e.g., [5]).

A very powerful possibility lies in the application of potential theoretical methods. By this means, the solutions for the subsurface stress fields have been obtained in closed

analytical form for the axisymmetric [2] and elliptical [6] frictionless Hertzian contact, as well as the axisymmetric [7,8] and elliptical [9] sliding Hertzian contact. Alternatively, at least for the axisymmetric frictionless normal contact, one can apply Hankel transforms, as pioneered by Sneddon [10]; based on this procedure, very recently the solution for the stress field beneath an axisymmetric punch with a profile in the form of a power law has been published [11], albeit in integral form.

To the author's knowledge, no other exact and explicit solutions have been published for the subsurface stress fields in elastic point contacts (for line contacts, the full stress field can be obtained easily based on Muskhelishvili's [12] potential). However, recently [13,14] it was suggested to calculate the subsurface stress state for elastic contacts with arbitrary axisymmetric convex profiles under normal and tangential load based on the superposition of rigid incremental translations of circular contact domains—an ingenious idea to solve axisymmetric contact problems, which stems from Mossakovski [15] and later Jäger [16], and which is described comprehensively in the handbook by Popov et al. [17]. This superposition, obviously, requires the full knowledge of the solution for a single (normal or tangential) rigid translation of a circular contact domain (for elastically decoupled problems, this corresponds to simple axisymmetric distributions of the normal or tangential contact stress, respectively). While the authors in [13] argue that the respective subsurface stress fields can be obtained as certain derivatives of the corresponding known solutions for a parabolic (i.e., Hertzian) contact, they fail to give the explicit solutions for the full stress fields they require for their superposition procedure (the aforementioned derivatives are extremely lengthy and impractical). Therefore, in the present manuscript, these stress fields shall be given in explicit closed form based on a potential theoretical approach.

The idea of superimposing incremental rigid translations of circular contact domains—and thus the exact solutions for the corresponding subsurface stress states presented in this manuscript—have a broad variety of numerical applications.

In numerical contact mechanics, there are methodically different approaches, which can be arranged in a spectrum ranging from "flexible but computationally costly, requiring little analytical preparatory work" to "specific but computationally simple, requiring much analytical preparatory work". Finite elements (FEs) [18] constitute the "flexible but computationally costly" end of that spectrum: an FE-based solution of a three-dimensional problem requires a three-dimensional discretization (with the respective demands of memory space and computational power); on the other hand, FEs have generally very few restrictions regarding physical modeling and require almost no analytical preparatory work.

An approach that is very common in numerical contact mechanics—and which has succesfully been applied for the calculation of subsurface stresses in frictional contacts already 30 years ago [19]—are (half-space) boundary elements (BEs) [20], usually accelerated by the fast Fourier transform (FFT). FFT-BE-based methods are in the middle of the aforementioned spectrum, as they only use a two-dimensional discretization (of the half-space surface), but are restricted to constitutionally linear problems and require the knowledge of the fundamental solutions for point loading on the surface.

With the superposition of incremental rigid translations of the contact domain, the discretization dimension is reduced to one (and, even more importantly, the degrees of freedom are decoupled)—characterizing the correct series of incremental translations to solve the contact problem—which allows for the real-time analysis of tribological contacts. However, it requires the knowledge of the analytical solution for a single such translation, which only is available if the problem obeys specific geometrical restrictions, e.g., axial symmetry. So, e.g., surface roughness cannot be considered. On the other hand, the superposition of incremental rigid translations is the basis of the method of dimensionality reduction (MDR, [17,21]), which has been succesfully applied to various frictional contact problems in engineering, also including, e.g., viscoelasticity or adhesion.

The remainder of the manuscript is organized as follows: In Section 2, the considered problem in linear elasticity is stated rigorously. Section 3 gives the solution for the stress field under normal loading of a circular surface domain, while Section 4 considers the

stress field under corresponding tangential loading. To illustrate how the obtained exact solutions for the subsurface stress field can be used to rapidly calculate subsurface stresses in general frictional contacts of axisymmetric, elastically similar bodies, in Section 5, the stress field under the tangential contact between a rigid cylindrical flat punch with rounded corners and an elastic half-space is considered in detail. Some conclusive remarks finish the manuscript.

## 2. Problem Statement

Let us consider a linearly elastic body that obeys the restrictions of the half-space approximation; the elastic material shall have the shear modulus $G$ and Poisson's ratio $\nu$ and occupy the half-space $z \geq 0$ in a Cartesian coordinate system $\{x, y, z\}$. Let there be loading in the form of normal and tangential tractions on a circular region with radius $a$ of the boundary surface $z = 0$ of the half-space. We will consider the following two loading scenarios: Firstly, there shall be loading in the form of a normal compressive stress according to the boundary conditions

$$
\begin{aligned}
\sigma_{yz}(z = 0) &= \sigma_{xz}(z = 0) = 0, \\
\sigma_{zz}(z = 0) &= -\frac{p_0 a}{\sqrt{a^2 - r^2}} \quad , \quad r < a.
\end{aligned}
\tag{1}
$$

Here, $\sigma_{jk}$ with $j, k = \{x, y, z\}$ are the components of the stress tensor, $r = \sqrt{x^2 + y^2}$ is the polar radius, and $p_0$ is a constant. On the other hand, let us analyze the case of uniaxial tangential tractions in the form

$$
\begin{aligned}
\sigma_{yz}(z = 0) &= \sigma_{zz}(z = 0) = 0, \\
\sigma_{xz}(z = 0) &= -\frac{\tau_0 a}{\sqrt{a^2 - r^2}} \quad , \quad r < a,
\end{aligned}
\tag{2}
$$

with a constant $\tau_0$.

While that is not too relevant for the solution to the thus stated problem in linear elasticity, to give some physical background, we shall briefly discuss, how the boundary conditions (1) and (2) could be realized in a mechanical contact problem.

If the elastic half-space is incompressible (i.e., $\nu = 0.5$), the boundary conditions are easily implemented by rigid translations of the circular contact domain ($z = 0 \wedge r \leq a$) in the $z$- and $x$-direction, respectively; in other words, by normal and tangential loading with a rigid cylindrical flat punch. However, for a compressible material, there will be elastic coupling between the normal and tangential contact problems, and the physical realization of the boundary conditions (1) and (2) will not be as straightforward as for the incompressible case.

Nonetheless, a common framework of analytical contact mechanics to solve contact problems with friction is the theory of Cattaneo [22] and Mindlin [23], which neglects (among other details) elastic coupling contributions. In [24], the Cattaneo–Mindlin approximate theory was compared to a rigorous numerical contact solution for the frictional Hertzian contact under shear load, and it was found, that the error of the approximation (in terms of, e.g., contact tractions) is generally small. Accordingly, the elasticity problems formulated in the above Equations (1) and (2) are a fundamental basis for the analysis of any tangential contact problem with friction of axisymmetric elastic bodies, as has been discussed in detail in the handbook [17] and as will be demonstrated briefly in Section 5.

## 3. Solution for the Normal Load

Let us start with the solution to the boundary value problem characterized by the boundary conditions (1). As the corresponding potential theoretical problem has been solved, and only the stress field needs to be computed from the known potential, this section can be kept short.

According to Barber ([25], pp. 64 f.), the stress and displacement fields in the elastic half-space can be expressed in terms of the axisymmetric harmonic potential

$$
\begin{aligned}
\varphi(r,z) &= -p_0 a \, \Re\left\{ \int_0^a F(r,z;\xi)\mathrm{d}\xi \right\}, \\
F(r,z;\xi) &= \ln\left( \sqrt{r^2 + [z + i\xi]^2} + z + i\xi \right),
\end{aligned}
\tag{3}
$$

with the imaginary unit $i = \sqrt{-1}$.

The stress field in (axisymmetric) cylindrical coordinates $\{r, \theta, z\}$ can be calculated from the potential based on the general relations ([25], p. 544)

$$
\begin{aligned}
\sigma_{rr} &= z\frac{\partial^3\varphi}{\partial r^2 \partial z} + \frac{\partial^2\varphi}{\partial r^2} - 2\nu\left( \frac{\partial^2\varphi}{\partial r^2} + \frac{\partial^2\varphi}{\partial z^2} \right), \\
\sigma_{\theta\theta} &= -(1 - 2\nu)\frac{\partial^2\varphi}{\partial r^2} - \frac{\partial^2\varphi}{\partial z^2} - z\frac{\partial^3\varphi}{\partial r^2 \partial z} - z\frac{\partial^3\varphi}{\partial z^3}, \\
\sigma_{rz} &= z\frac{\partial^3\varphi}{\partial r \partial z^2}, \\
\sigma_{zz} &= z\frac{\partial^3\varphi}{\partial z^3} - \frac{\partial^2\varphi}{\partial z^2}.
\end{aligned}
\tag{4}
$$

All other stress components vanish because of the problem's symmetry.

Moreover, for the non-vanishing components of the displacement field, we have ([25], p. 544)

$$
\begin{aligned}
2Gu_r &= z\frac{\partial^2\varphi}{\partial r \partial z} + (1 - 2\nu)\frac{\partial\varphi}{\partial r}, \\
2Gu_z &= z\frac{\partial^2\varphi}{\partial z^2} - 2(1 - \nu)\frac{\partial\varphi}{\partial z}.
\end{aligned}
\tag{5}
$$

Putting the solution for the elastic potential (3) into the general relations (4), and executing the resulting derivatives and integrals does not pose mathematical difficulties. The resulting stress field is given by the imaginary parts of the complex-valued field

$$
\begin{aligned}
\frac{\hat{\sigma}_{rr}(r,z)}{p_0 a} &= -\frac{z}{r^2}\frac{u(2r^2 + u^2)}{(r^2 + u^2)^{3/2}} + \frac{1}{\sqrt{r^2 + u^2}} + \frac{1 - 2\nu}{r^2}\left( u - \sqrt{r^2 + u^2} \right), \\
\frac{\hat{\sigma}_{rz}(r,z)}{p_0 a} &= \frac{rz}{(r^2 + u^2)^{3/2}}, \\
\frac{\hat{\sigma}_{zz}(r,z)}{p_0 a} &= \frac{uz}{(r^2 + u^2)^{3/2}} + \frac{1}{\sqrt{r^2 + u^2}}, \\
\frac{\hat{\sigma}_{\theta\theta}(r,z)}{p_0 a} &= \frac{2\nu}{\sqrt{r^2 + u^2}} + \frac{uz}{r^2\sqrt{r^2 + u^2}} - \frac{1 - 2\nu}{r^2}\left( u - \sqrt{r^2 + u^2} \right),
\end{aligned}
\tag{6}
$$

with the complex auxiliary variable

$$
u = z + ia.
\tag{7}
$$

In Figure 1, the radial and circumferential components of the physical stress field are shown as contour line diagrams, in normalized variables, with $\nu = 0.3$. It is visible that the surface of the half-space, outside the contact circle, experiences tensile stresses in the radial direction. On the other hand, the circumferential normal stresses are compressive everywhere.

Figure 2 gives the corresponding contour line diagrams of the $\sigma_{zz}$ stress component and the Von Mises equivalent stress, in the same normalized variables.

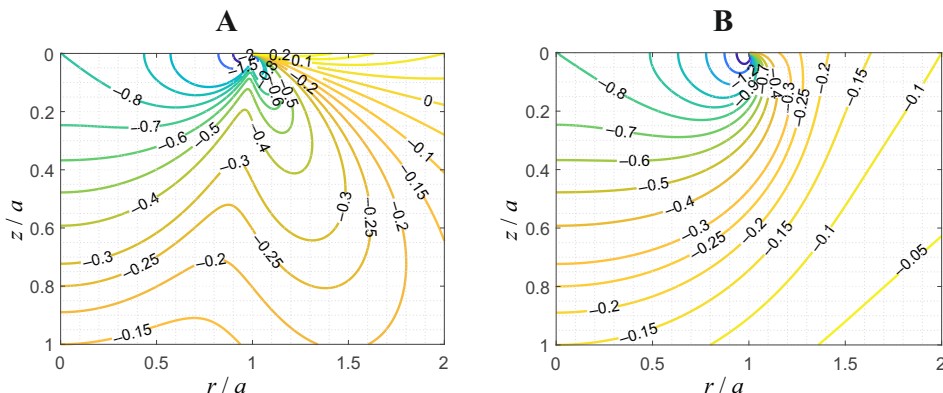

**Figure 1.** Contour line diagrams of the radial component $\sigma_{rr}$ (**A**) and the circumferential component $\sigma_{\theta\theta}$ (**B**) of the subsurface stress field due to the normal loading (1), normalized for $p_0$, with $\nu = 0.3$.

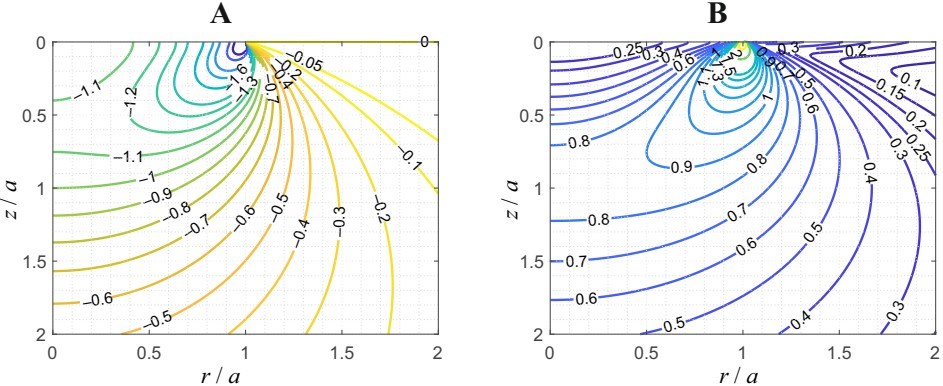

**Figure 2.** Contour line diagrams of the vertical stress component $\sigma_{zz}$ (**A**) and the Von Mises equivalent stress (**B**) of the subsurface stress field due to the normal loading (1), normalized for $p_0$, with $\nu = 0.3$.

Similarly, we can use the potential (3) in the general relations (5) to obtain the displacement field as the imaginary parts of the complex-valued field

$$
\begin{aligned}
\frac{2G\hat{u}_r}{p_0 a} &= \frac{z}{r}\frac{u}{\sqrt{r^2 + u^2}} + \frac{1 - 2\nu}{r}\left(\sqrt{r^2 + u^2} - u\right), \\
\frac{2G\hat{u}_z}{p_0 a} &= -\frac{z}{\sqrt{r^2 + u^2}} + 2(1 - \nu)\ln\left(\frac{\sqrt{r^2 + u^2} + u}{a}\right)
\end{aligned}
\tag{8}
$$

In the surface $z = 0$, the corresponding imaginary parts of the complex-valued field (6) can be explicitly evaluated easily. We obtain for the stress state in the surface inside the contact domain

$$
\begin{aligned}
\frac{\sigma_{rr}(r < a, z = 0)}{p_0 a} &= -\frac{1}{\sqrt{a^2 - r^2}} + \frac{1 - 2\nu}{r^2}\left(a - \sqrt{a^2 - r^2}\right), \\
\frac{\sigma_{zz}(r < a, z = 0)}{p_0 a} &= -\frac{1}{\sqrt{a^2 - r^2}}, \\
\frac{\sigma_{\theta\theta}(r < a, z = 0)}{p_0 a} &= -\frac{2\nu}{\sqrt{a^2 - r^2}} - \frac{1 - 2\nu}{r^2}\left(a - \sqrt{a^2 - r^2}\right),
\end{aligned}
\tag{9}
$$

and outside the contact domain

$$
\frac{\sigma_{rr}(r > a, z = 0)}{p_0 a} = -\frac{\sigma_{\theta\theta}(r > a, z = 0)}{p_0 a} = \frac{(1 - 2\nu)a}{r^2}.
\tag{10}
$$

All other stress components vanish. The results (9) and (10) are in perfect agreement with the ones reported in [13] for the same problem.

On the axis of symmetry, $r = 0$, the stresses have to be reworked. For the non-vanishing physical stresses one obtains

$$
\begin{aligned}
\frac{\sigma_{rr}(r=0,z)}{p_0 a} &= \frac{\sigma_{\theta\theta}(r=0,z)}{p_0 a} = -\frac{a^3}{(a^2+z^2)^2} + (1-2\nu)\frac{a}{2(a^2+z^2)}, \\
\frac{\sigma_{zz}(r=0,z)}{p_0 a} &= -\frac{a^3 + 3az^2}{(a^2+z^2)^2}.
\end{aligned} \tag{11}
$$

While the above formulation of the stress field in axisymmetric cylindrical coordinates is very compact, it sometimes may be preferable to have the field in the original Cartesian coordinate system, especially if one is interested in stress states resulting from superimposed normal and tangential loading because the contact under tangential load looses its rotational symmetry.

Based on the transformation rules for the stress tensor between cartesian and axisymmetric cylindrical coordinates, we obtain the stress field as the imaginary parts of

$$
\begin{aligned}
\frac{\hat{\sigma}_{xx}}{p_0 a} &= \frac{uz}{r^4}\left[\frac{y^2}{\sqrt{r^2+u^2}} - \frac{x^2(2r^2+u^2)}{(r^2+u^2)^{3/2}}\right] + \frac{x^2+2\nu y^2}{r^2\sqrt{r^2+u^2}} + \frac{x^2-y^2}{r^4}(1-2\nu)\left(u - \sqrt{r^2+u^2}\right), \\
\frac{\hat{\sigma}_{yy}}{p_0 a} &= \frac{uz}{r^4}\left[\frac{x^2}{\sqrt{r^2+u^2}} - \frac{y^2(2r^2+u^2)}{(r^2+u^2)^{3/2}}\right] + \frac{y^2+2\nu x^2}{r^2\sqrt{r^2+u^2}} + \frac{y^2-x^2}{r^4}(1-2\nu)\left(u - \sqrt{r^2+u^2}\right), \\
\frac{\hat{\sigma}_{xy}}{p_0 a} &= \frac{xy}{r^2}\left\{-\frac{uz}{r^2}\frac{3r^2+2u^2}{(r^2+u^2)^{3/2}} + (1-2\nu)\left[\frac{1}{\sqrt{r^2+u^2}} + \frac{2}{r^2}\left(u - \sqrt{r^2+u^2}\right)\right]\right\}, \\
\frac{\hat{\sigma}_{xz}}{p_0 a} &= \frac{xz}{(r^2+u^2)^{3/2}}, \\
\frac{\hat{\sigma}_{yz}}{p_0 a} &= \frac{yz}{(r^2+u^2)^{3/2}}.
\end{aligned} \tag{12}
$$

In the surface inside the contact domain, the physical stresses simplify to

$$
\begin{aligned}
\frac{\sigma_{xx}(r<a,z=0)}{p_0 a} &= -\frac{x^2+2\nu y^2}{r^2\sqrt{a^2-r^2}} + \frac{x^2-y^2}{r^4}(1-2\nu)\left(a - \sqrt{a^2-r^2}\right), \\
\frac{\sigma_{yy}(r<a,z=0)}{p_0 a} &= -\frac{y^2+2\nu x^2}{r^2\sqrt{a^2-r^2}} + \frac{y^2-x^2}{r^4}(1-2\nu)\left(a - \sqrt{a^2-r^2}\right), \\
\frac{\sigma_{xy}(r<a,z=0)}{p_0 a} &= \frac{xy}{r^2}(1-2\nu)\left[-\frac{1}{\sqrt{a^2-r^2}} + \frac{2}{r^2}\left(a - \sqrt{a^2-r^2}\right)\right],
\end{aligned} \tag{13}
$$

while in the surface, but outside the contact domain

$$
\begin{aligned}
\frac{\sigma_{xx}(r>a,z=0)}{p_0 a} &= \frac{(x^2-y^2)a}{r^4}(1-2\nu), \\
\frac{\sigma_{yy}(r>a,z=0)}{p_0 a} &= \frac{(y^2-x^2)a}{r^4}(1-2\nu), \\
\frac{\sigma_{xy}(r>a,z=0)}{p_0 a} &= \frac{2xya}{r^4}(1-2\nu).
\end{aligned} \tag{14}
$$

On the axis of symmetry, we obtain from Equations (11)

$$
\frac{\sigma_{xx}(r=0,z)}{p_0 a} = \frac{\sigma_{yy}(r=0,z)}{p_0 a} = -\frac{a^3}{(a^2+z^2)^2} + (1-2\nu)\frac{a}{2(a^2+z^2)}. \tag{15}
$$

## 4. Solution for the Tangential Load

We now turn our attention to the boundary value problem characterized by the boundary conditions (2).

For the solution of the potential theoretical problem, we will apply the procedure suggested by Hamilton and Goodman [7] for the sliding circular Hertzian contact.

According to [7], the displacement field can be written in terms of a harmonic stress function $T$ as follows:

$$
\begin{aligned}
2Gu_x &= 2\nu\frac{\partial^2 T}{\partial x^2} + 2\frac{\partial^2 T}{\partial z^2} - z\frac{\partial^3 T}{\partial x^2 \partial z}, \\
2Gu_y &= 2\nu\frac{\partial^2 T}{\partial x \partial y} - z\frac{\partial^3 T}{\partial x \partial y \partial z}, \\
2Gu_z &= (1-2\nu)\frac{\partial^2 T}{\partial x \partial z} - z\frac{\partial^3 T}{\partial x \partial z^2}.
\end{aligned}
\tag{16}
$$

Applying Hooke's law and accounting for the fact that $T$ is harmonic, one obtains the stresses in terms of the potential,

$$
\begin{aligned}
\sigma_{xy} &= 2\nu\frac{\partial^3 T}{\partial x^2 \partial y} + \frac{\partial^3 T}{\partial y \partial z^2} - z\frac{\partial^4 T}{\partial x^2 \partial y \partial z}, \\
\sigma_{xz} &= \frac{\partial^3 T}{\partial z^3} - z\frac{\partial^4 T}{\partial x^2 \partial z^2}, \\
\sigma_{yz} &= -z\frac{\partial^4 T}{\partial x \partial y \partial z^2}, \\
\sigma_{xx} &= 2\nu\frac{\partial^3 T}{\partial x^3} + 2(1+\nu)\frac{\partial^3 T}{\partial x \partial z^2} - z\frac{\partial^4 T}{\partial x^3 \partial z}, \\
\sigma_{yy} &= -2\nu\frac{\partial^3 T}{\partial x^3} - z\frac{\partial^4 T}{\partial x \partial y^2 \partial z}, \\
\sigma_{zz} &= -z\frac{\partial^4 T}{\partial x \partial z^3}.
\end{aligned}
\tag{17}
$$

The potential $T$ is given by the imaginary part of [7]

$$
\hat{T} = \int_0^a t(\xi)\left[\frac{1}{2}\left(z_1^2 - \frac{r^2}{2}\right)\ln\left(z_1 + \sqrt{r^2 + z_1^2}\right) - \frac{3}{4}z_1\sqrt{r^2 + z_1^2} + \frac{r^2}{4}\right]\mathrm{d}\xi,
\tag{18}
$$

with the complex auxiliary variable $z_1 = z + i\xi$, and a weight function $t(\xi)$ that is yet to be determined from the boundary conditions at the half-space surface. Putting the potential (18) into the general relations (17) and comparing with the boundary conditions (2) (there seems to be missing a minus sign in Equation (5) of [7]), one determines that the correct weight function $t(\xi)$ is actually the Dirac distribution, and the harmonic potential which satisfies the boundary condition (2) is therefore given by

$$
T(r, u) = \tau_0 a\,\mathfrak{Im}\left\{\frac{1}{2}\left(u^2 - \frac{r^2}{2}\right)\ln\left(u + \sqrt{r^2 + u^2}\right) - \frac{3}{4}u\sqrt{r^2 + u^2} + \frac{r^2}{4}\right\},
\tag{19}
$$

with the complex auxiliary variable $u$ as in Equation (7).

The derivatives in Equations (16) and (17) can be evaluated without severe difficulties. The resulting stress field is given by the imaginary parts of the complex-valued field

$$\frac{\hat{\sigma}_{xy}}{\tau_0 a} = 2\nu\left[\frac{y}{r^2}\left(\frac{u}{\rho+u}-\frac{1}{2}\right)+\frac{x^2 y}{\rho(\rho+u)^3}\right]+\frac{y}{\rho(\rho+u)}-z\left[\frac{y}{\rho(\rho+u)^2}-\frac{x^2 y(3\rho+u)}{\rho^3(\rho+u)^3}\right],$$

$$\frac{\hat{\sigma}_{xz}}{\tau_0 a} = \frac{1}{\rho}-z\left[\frac{1}{\rho(\rho+u)}-\frac{x^2(2\rho+u)}{\rho^3(\rho+u)^2}\right],$$

$$\frac{\hat{\sigma}_{yz}}{\tau_0 a} = \frac{xyz(2\rho+u)}{\rho^3(\rho+u)^2},$$

$$\frac{\hat{\sigma}_{xx}}{\tau_0 a} = 2\nu\left[\frac{3x}{r^2}\left(\frac{u}{\rho+u}-\frac{1}{2}\right)+\frac{x^3}{\rho(\rho+u)^3}\right]+\frac{2x(1+\nu)}{\rho(\rho+u)}-z\left[\frac{3x}{\rho(\rho+u)^2}-\frac{x^3(3\rho+u)}{\rho^3(\rho+u)^3}\right],$$ (20)

$$\frac{\hat{\sigma}_{yy}}{\tau_0 a} = -2\nu\left[\frac{3x}{r^2}\left(\frac{u}{\rho+u}-\frac{1}{2}\right)+\frac{x^3}{\rho(\rho+u)^3}\right]-z\left[\frac{x}{\rho(\rho+u)^2}-\frac{xy^2(3\rho+u)}{\rho^3(\rho+u)^3}\right],$$

$$\frac{\hat{\sigma}_{zz}}{\tau_0 a} = \frac{xz}{\rho^3},$$

with the complex auxiliary variable

$$\rho = \sqrt{r^2+u^2} = \sqrt{r^2+z^2-a^2+2iaz},$$ (21)

and the displacement field is given by the imaginary parts of the complex-valued field

$$\frac{2G\hat{u}_x}{\tau_0 a} = \frac{\nu}{2}+(2-\nu)\ln\left(\frac{\rho+u}{a}\right)+z\left[\frac{1}{\rho+u}-\frac{x^2}{\rho(\rho+u)^2}\right]+2\nu\left[\frac{u}{2r^2}(u-\rho)+\frac{x^2}{r^2}\left(\frac{u}{\rho+u}-\frac{1}{2}\right)\right],$$

$$\frac{2G\hat{u}_y}{\tau_0 a} = \frac{2\nu xy}{r^2}\left(\frac{u}{\rho+u}-\frac{1}{2}\right)-\frac{xyz}{\rho(\rho+u)^2},$$ (22)

$$\frac{2G\hat{u}_z}{\tau_0 a} = -x\left[\frac{1-2\nu}{\rho+u}+\frac{z}{\rho(\rho+u)}\right].$$

As the fields depend on all three cartesian coordinates, it is difficult to comprehensively visualize all dependencies.

In Figure 3, contour line diagrams of the $\sigma_{xx}$ and $\sigma_{xz}$ stress components are shown in the plane $y = 0$ and in normalized variables. Because of the problem's symmetry, it is clear that $\sigma_{xx}$ is antisymmetric in $x$, while $\sigma_{xz}$ is symmetric in $x$.

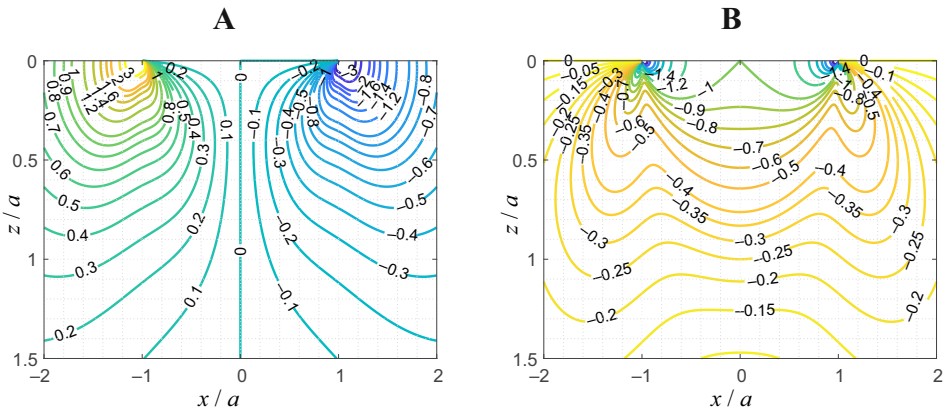

**Figure 3.** Contour line diagrams of the normal stress component $\sigma_{xx}$ (**A**) and the tangential stress component $\sigma_{xz}$ (**B**) of the subsurface stress field in the *x-z* plane ($y = 0$) due to the tangential loading (2), normalized for $\tau_0$, with $\nu = 0.3$.

Figure 4 gives the corresponding contour line diagrams of the $\sigma_{zz}$ stress component and the Von Mises equivalent stress, in the same normalized variables. The distribution of

the equivalent stress is very similar to the one of the $\sigma_{xz}$ stress component. So, we conclude that the Von Mises equivalent stress is dominated by this shear stress.

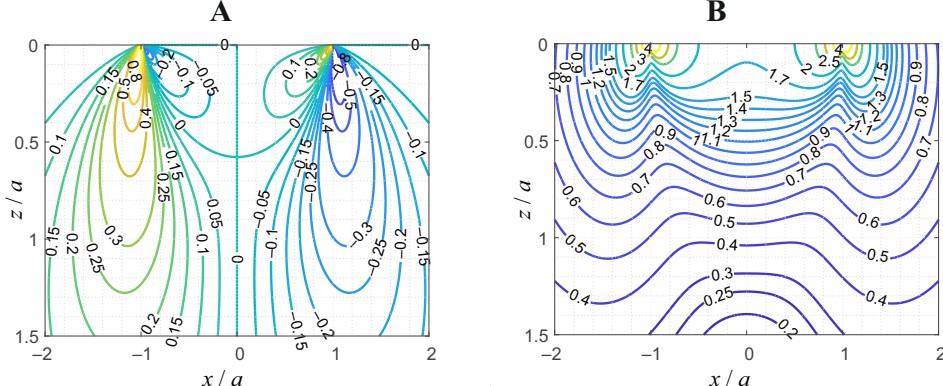

**Figure 4.** Contour line diagrams of the vertical stress component $\sigma_{zz}$ (**A**) and the equivalent Von Mises stress (**B**) of the subsurface stress field in the *x-z* plane ($y = 0$) due to the tangential loading (2), normalized for $\tau_0$, with $\nu = 0.3$.

To demonstrate the variation of the stress field in the *y*-direction, in Figures 5 and 6, contour line diagrams are shown of all six stress components in the subsurface plane $z = a$, as functions of $x$ and $y$ and in normalized variables.

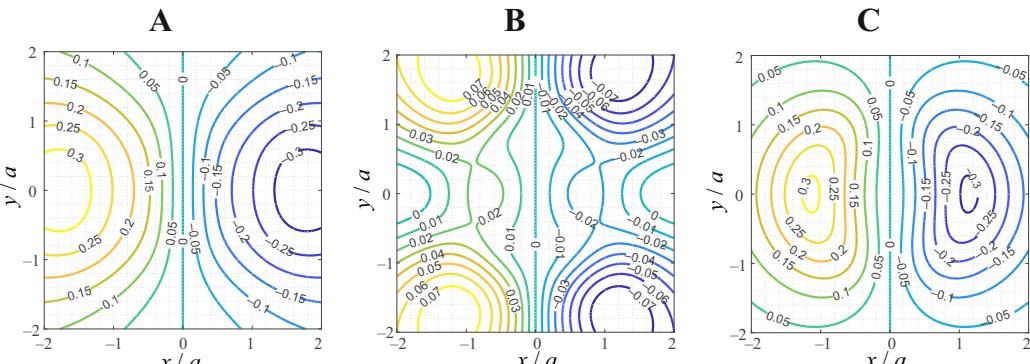

**Figure 5.** Contour line diagrams of the normal stress components $\sigma_{xx}$ (**A**), $\sigma_{yy}$ (**B**), and $\sigma_{zz}$ (**C**) of the subsurface stress field in the plane $z = a$ due to the tangential loading (2), normalized for $\tau_0$, with $\nu = 0.3$.

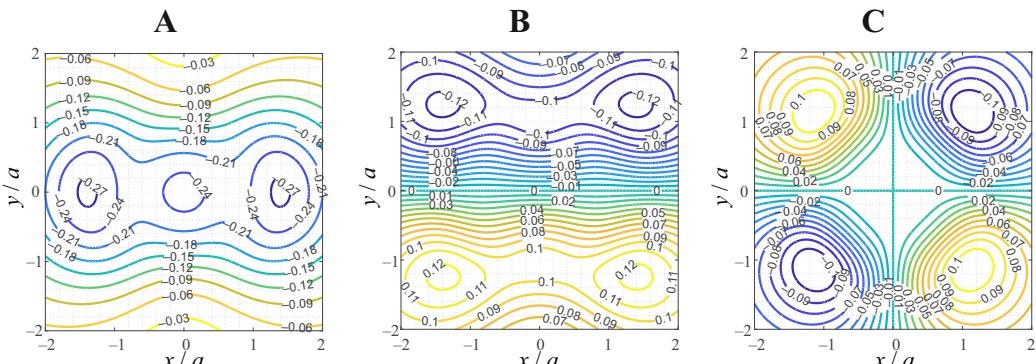

**Figure 6.** Contour line diagrams of the shear stress components $\sigma_{xz}$ (**A**), $\sigma_{xy}$ (**B**), and $\sigma_{yz}$ (**C**) of the subsurface stress field in the plane $z = a$ due to the tangential loading (2), normalized for $\tau_0$, with $\nu = 0.3$.

In the surface, inside the contact domain, the only non-vanishing physical stress component is

$$\frac{\sigma_{xz}(r < a, z = 0)}{\tau_0 a} = -\frac{1}{\sqrt{a^2 - r^2}}. \tag{23}$$

In the surface, but outside the contact domain, the non-vanishing physical stress components are

$$
\begin{aligned}
\frac{\sigma_{xy}(r > a, z = 0)}{\tau_0 a} &= -\frac{ya}{r^4}\left[\frac{r^2}{\sqrt{r^2 - a^2}} - 2v\left\{\left(1 - \frac{4x^2}{r^2}\right)\sqrt{r^2 - a^2} + \frac{x^2}{\sqrt{r^2 - a^2}}\right\}\right], \\
\frac{\sigma_{xx}(r > a, z = 0)}{\tau_0 a} &= -\frac{xa}{r^4}\left[\frac{2r^2}{\sqrt{r^2 - a^2}} - 2v\left\{\left(3 - \frac{4x^2}{r^2}\right)\sqrt{r^2 - a^2} - \frac{y^2}{\sqrt{r^2 - a^2}}\right\}\right], \\
\frac{\sigma_{yy}(r > a, z = 0)}{\tau_0 a} &= -2v\frac{xa}{r^4}\left[\left(3 - \frac{4x^2}{r^2}\right)\sqrt{r^2 - a^2} + \frac{x^2}{\sqrt{r^2 - a^2}}\right],
\end{aligned}
\tag{24}
$$

which agrees with the results reported in [13] for the same problem.

On the *z*-axis, the only non-vanishing physical stress component is

$$\frac{\sigma_{xz}(r = 0, z)}{\tau_0 a} = -\frac{a^3}{(a^2 + z^2)^2}. \tag{25}$$

## 5. Application: Subsurface Stress Field in Frictional Contacts of Elastically Similar Axisymmetric Bodies

As was mentioned before, the exact solutions derived in the previous two sections can be used to very rapidly compute the subsurface stress fields in general frictional contacts of axisymmetric bodies in a semi-analytic fashion. Based on the obtained analytical results for the subsurface elastic stress state under rigid translations of a circular surface domain, the numerical effort for the determination of the subsurface stresses in general frictional elastic contacts of (elastically similar) axisymmetric bodies is reduced to the evaluation of elementary one-dimensional integrals. In the present section, the corresponding semi-analytical procedure shall be described and illustrated.

### 5.1. General Procedure for Arbitrary Convex Profile Geometries

For this purpose, let us consider two linearly elastic, homogeneous, isotropic bodies—which obey the restrictions of the half-space approximation—with the shear moduli $G_1$ and $G_2$ and Poisson's ratios $v_1$ and $v_2$. The materials shall be elastically similar to avoid elastic coupling between the normal and tangential contact problems, i.e.,

$$\frac{1 - 2v_1}{G_1} = \frac{1 - 2v_2}{G_2}. \tag{26}$$

Moreover, let the gap between the contacting surfaces, at the moment of first contact, be an axisymmetric smooth monotonous function $f = f(r)$.

If the two bodies are in contact over a circular domain with radius $\tilde{a}$, and two remote points of the bodies on the axis of symmetry are moved to one another by an incremental indentation depth $dd$, the bodies will experience incremental surface tractions ([17], p. 12)

$$
\begin{aligned}
\sigma_{yz}(z = 0) &= \sigma_{xz}(z = 0) = 0, \\
d\sigma_{zz}(z = 0) &= -\frac{E^*}{\pi}\frac{dd}{\sqrt{\tilde{a}^2 - r^2}} \quad, \quad r < \tilde{a},
\end{aligned}
\tag{27}
$$

with the effective Young's modulus

$$\frac{1}{E^*} = \frac{1 - v_1}{2G_1} + \frac{1 - v_2}{2G_2}. \tag{28}$$

Obviously, Equations (27) are of the same form as Equations (1).

Similarly, if two remote points on the axis of symmetry are moved relative to one another in the tangential direction by an incremental displacement $du_0$ (without slip and without tilting), the bodies will experience incremental surface tractions ([17], p. 137)

$$
\begin{aligned}
\sigma_{yz}(z=0) &= \sigma_{zz}(z=0) = 0, \\
d\sigma_{xz}(z=0) &= -\frac{G^*}{\pi}\frac{du_0}{\sqrt{\tilde{a}^2 - r^2}} \quad, \quad r < \tilde{a},
\end{aligned}
\tag{29}
$$

with the effective shear modulus

$$
\frac{1}{G^*} = \frac{2-\nu_1}{4G_1} + \frac{2-\nu_2}{4G_2}.
\tag{30}
$$

Once again, Equations (29) are of the same form as Equations (2).

Now, let us consider the full normal indentation procedure, i.e., from the indentation depth $\tilde{d} = 0$ to the final indentation depth $\tilde{d} = d$. The indentation depth is a unique function of the contact radius ([17], p. 10),

$$
\tilde{d} = g(\tilde{a}) = \tilde{a} \int_0^{\tilde{a}} \frac{f'(r)\,dr}{\sqrt{\tilde{a}^2 - r^2}},
\tag{31}
$$

with the prime denoting the first derivative, and accordingly,

$$
d\tilde{d} = g'(\tilde{a})d\tilde{a}.
\tag{32}
$$

As Equations (1) and (27) have exactly the same form, the subsurface stress state for the axisymmetric normal contact problem can be superimposed as

$$
\sigma_{jk}^{\text{norm}}(x,y,z;a) = \frac{E^*}{\pi} \int_0^a \frac{\sigma_{jk}^{(1)}(x,y,z;\tilde{a})}{p_0\tilde{a}} g'(\tilde{a})\,d\tilde{a},
\tag{33}
$$

where $\sigma_{jk}^{(1)}$ denotes the stress field due to the surface loading (1), which has been given explicitly in Section 3.

When numerically evaluating the integral (33), a little care is necessary due to the stress singularity of $\sigma_{jk}^{(1)}$ at the edge of the contact domain. From Equation (9) it is clear that the singular contributions to the field all have the form

$$
\sigma_{jk}^{(1)}(z=0, r \to \tilde{a}^-) = -K_{jk}(x,y)\frac{p_0\tilde{a}}{\sqrt{\tilde{a}^2 - r^2}},
\tag{34}
$$

with different forms for $K_{jk}(x,y)$ for the different stress components. All other stress contributions are non-singular.

For the superposition integral (33), following Benad [26], we can then integrate by parts,

$$
\int_r^a \frac{g'(\tilde{a})}{\sqrt{\tilde{a}^2 - r^2}}\,d\tilde{a} = \cosh^{-1}\left(\frac{a}{r}\right)g'(a) - \int_r^a \cosh^{-1}\left(\frac{\tilde{a}}{r}\right)g''(\tilde{a})\,d\tilde{a},
\tag{35}
$$

with the area hyperbolic cosine, $\cosh^{-1}$, for optimal numerical performance.

Similarly, the subsurface stress field due to tangential loading can be sumperimposed as

$$
\sigma_{jk}^{\text{tang}}(x,y,z;a) = \frac{G^*}{\pi} \int_0^a \frac{\sigma_{jk}^{(2)}(x,y,z;\tilde{a})}{\tau_0\tilde{a}} u_0'(\tilde{a})d\tilde{a},
\tag{36}
$$

where $\sigma_{jk}^{(2)}$ denotes the stress field due to the surface loading (2), which has been given explicitly in Section 4. The singularities in the surface fields (24) outside the contact area, can be avoided for the stress superposition (36) using again integration by parts,

$$\int_0^r \frac{\tilde{a}}{\sqrt{r^2 - \tilde{a}^2}} \, u_0'(\tilde{a}) \, \mathrm{d}\tilde{a} = r u_0'(0) + \int_0^r \sqrt{r^2 - \tilde{a}^2} \, u_0''(\tilde{a}) \, \mathrm{d}\tilde{a}. \tag{37}$$

However, what is the tangential displacement "history" $u_0(\tilde{a})$, which generates the correct contact configuration? Consider Cattaneo's problem of a constant normal load and a subsequently applied monotonously increasing tangential load: The contact domain (within the approximation of Cattaneo and Mindlin, who neglect the lateral surface displacements $u_y$ for the contact solution) generally consists of an inner stick area with radius $c$ and an annulus $c < r \leq a$ of local slip. In the stick region, the tangential surface displacements must be constant, and in the slip annulus the surface tractions are connected by Amonton's law $|\sigma_{xz}(z = 0, c < r \leq a)| = \mu |\sigma_{zz}(z = 0, c < r \leq a)|$, with the coefficient of friction $\mu$.

Now consider the following series of incremental rigid translations. Up to a contact radius $\tilde{a} = c$ no displacement $u_0$ is imposed. After that, up to the final contact radius $\tilde{a} = a$, any incremental normal translation $\mathrm{d}d$ is accompanied by a tangential translation

$$\mathrm{d}u_0 = \frac{E^*}{G^*} \mu \mathrm{d}d. \tag{38}$$

The resulting contact configuration satisfies all boundary conditions of the Cattaneo–Mindlin problem ([17], p. 333). Hence, the function $u_0(\tilde{a})$ for Cattaneo's problem reads

$$u_0^{\mathrm{CM}}(\tilde{a}) = \frac{\mu E^*}{G^*} [g(\tilde{a}) - g(c)] H(\tilde{a} - c), \tag{39}$$

with the Heaviside step function $H$.

For more general loading histories, $u_0$ can either be superimposed based on Jäger's algorithm [27] or via the method of dimensionality reduction (MDR) [21].

For contacts with superimposed normal and tangential loading, of course, the stress states (33) and (36) can be simply added up. The procedure for the determination of the subsurface stress state in elastic frictional contacts of (elastically similar) convex, axisymmetric bodies can thus be summarized as follows:

1. Determine the auxiliary function $g$ from the gap function $f$, based on Equation (31).
2. Determine the subsurface stresses resulting from the normal loading, based on the superposition integral (33).
3. Determine the series of rigid tangential translations which reproduces the tangential contact configuration. For the Cattaneo–Mindlin loading history, Equation (39) can be used; for more general oblique loading, an MDR contact solver should be implemented.
4. Determine the subsurface stresses resulting from the tangential loading, based on the superposition integral (36).

*5.2. Example: Contact with a Cylindrical Flat Punch with Rounded Corners*

As an illustrative example, let us consider the axisymmetric profile

$$f(r) = \frac{(r - b)^2}{2R} H(r - b), \tag{40}$$

with two radii $b$ and $R$. As the elastic bodies shall be elastically similar, the tangential contact can be reduced to the one between an elastic half-space with the effective moduli $E^*$ and $G^*$, and a rigid counterbody with the profile $f$; the profile (40) corresponds to a cylindrical flat punch with rounded corners, as shown in Figure 7. The radius of the flat face of the punch is $b$, and the curvature radius of the rounded corners $R$.

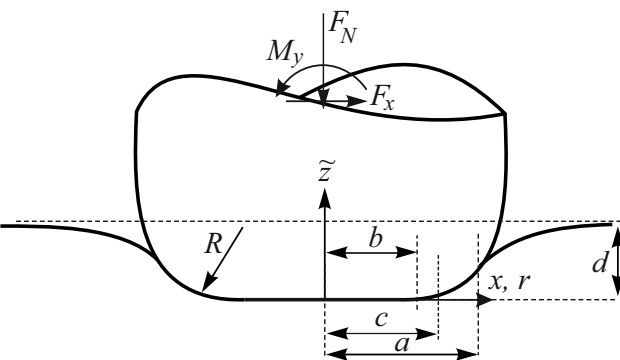

**Figure 7.** Tangential contact between an elastic half-space and a rigid cylindrical flat punch with rounded corners; notations are explained in the text.

The procedure summarized at the end of the previous subsection can be executed without problems. For the auxiliary function $g$, we have ([17], p. 41)

$$g(\tilde{a}) = \frac{\tilde{a}}{R}\left[\sqrt{\tilde{a}^2 - b^2} - b\arccos\left(\frac{b}{\tilde{a}}\right)\right]H(\tilde{a} - b). \tag{41}$$

Figure 8 shows the equivalent Von Mises stress in the $x$-$z$ plane (for $y = 0$) for the frictionless normal contact and the sliding tangential contact (with $\mu = 0.3$) in normalized variables. As the coefficient of friction is still relatively small (compared to 1), the stress field is only slightly altered by the sliding tractions. Notably, the maximum of the equivalent stress moves to the surface.

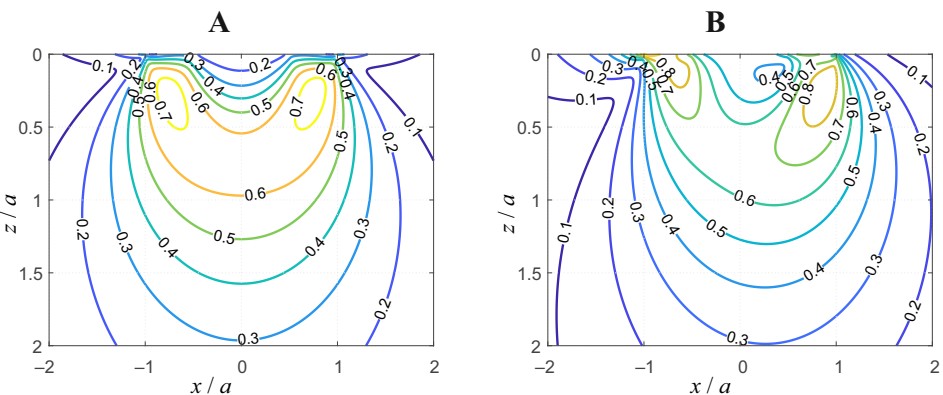

**Figure 8.** Contour line diagrams of the equivalent Von Mises stress in the $x$-$z$ plane ($y = 0$) for the frictionless normal contact (**A**) and sliding tangential contact with $\mu = 0.3$ (**B**) for the contact problem shown in Figure 7, normalized with the average contact pressure, with $\nu = 0.3$ and $b/a = 0.5$.

It may be interesting to analyze, how, for the sliding contact, the maximum of the Von Mises equivalent stress increases and moves towards the surface for increasing values of the coefficient of friction $\mu$. This is demonstrated in Figure 9, for the sliding contact with a rounded cylindrical flat punch, with $\nu = 0.3$ and $b/a = 0.5$ (the maximum of the equivalent stress always is in the plane with $y = 0$). As is already clear from Figure 8, there are two local maxima of the equivalent stress for the sliding contact: one below the surface at the leading edge (with $x > 0$), and one actually exactly at the trailing contact edge ($z = 0$ and $x = -a$); in Figure 9, for $\mu > 0.21$, the second maximum overpowers the first one, and for greater values of the coefficient of friction, the maximum equivalent stress increases linearly with the friction coefficient.

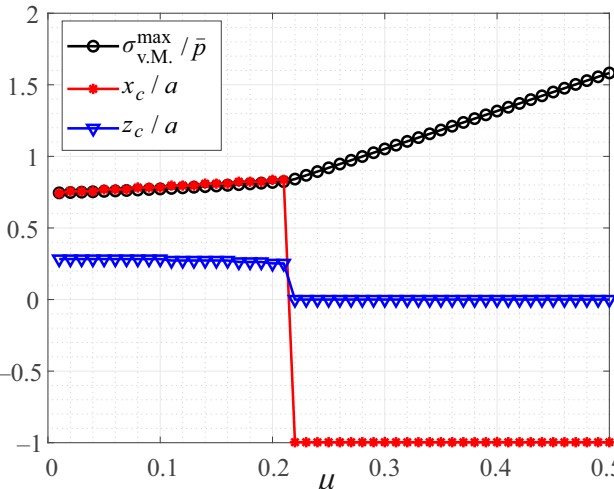

**Figure 9.** Maximum of the Von Mises equivalent stress, normalized for the average contact pressure, and its normalized location $(x_c, z_c)$, as a function of the coefficient of friction $\mu$, for the sliding tangential contact with a rounded cylindrical flat punch, with $\nu = 0.3$ and $b/a = 0.5$.

## 6. Discussion and Conclusions

Based on known potential theoretical procedures, the subsurface stress fields have been calculated exactly for an elastic half-space, which is subject to surface tractions that—in the case of elastic decoupling—correspond to rigid normal and tangential translations of a circular surface domain.

Within the framework of the Cattaneo–Mindlin approximation, any tangential frictional contact problem of convex, axisymmetric elastic bodies can be solved as a specific series of such (incremental) rigid translations [17]. In this sense, the presented solutions allow for a very fast calculation of subsurface stress fields for arbitrary axisymmetric elastic tangential contacts with friction. This can be used, e.g., for the real-time analysis of tribological contacts or for large parameter studies with respect to subsurface yield or fatigue cracking under dynamic loads [28], like impacts or fretting oscillations.

It should be noted that the obtained solutions are mathematically exact within the restrictions of the physical modeling—static, linearly elastic deformation of a half-space—and can thus also serve as benchmark solutions for different numerical contact algorithms.

On the other hand, it is clear that the underlying physical modeling of the problem formulation poses restrictions that have to be kept in mind when applying the shown solutions to real engineering contacts. Also, surface roughness is not considered.

For future work, it may be desirable to account for elastic coupling—at least, in the framework of the Goodman approximation [29]—or for prestresses in the elastic half-space ([25], p. 179) due to, e.g., external loading of the contacting bodies far from the contact.

Based on the correspondence between boundary values problems in linear elasticity and linear viscoelasticity, the shown solutions and procedures can also be applied to viscoelastic contacts [30].

**Funding:** This research was funded by the German Research Foundation under the project number PO 810/66-1. We acknowledge support by the Open Access Publication Fund of TU Berlin.

**Data Availability Statement:** Not applicable.

**Conflicts of Interest:** The author declares no conflicts of interest.

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
