# Peer review of "Elastic Stress Field beneath a Sticking Circular Contact under Tangential Load"

_solids, doi:10.3390/solids5010002_

Round 1

Reviewer 1 Report

Comments and Suggestions for Authors

The novelty of the paper is not clear. The author needs to clarify the novelty of this work. The numerical solutions of subsurface stress field including tangential shear already exist e.g.,

1. Ali, F., 2019. Numerical study on subsurface stress in Hertzian contacts under pure sliding conditions. Journal of Applied and Computational Mechanics.

2. Lee, S.C. and Ren, N., 1994. The subsurface stress field created by three-dimensionally rough bodies in contact with traction. Tribology transactions, 37(3), pp.615-621.

Some of the analytical solutions are also available in the relevant textbooks such as Johnson’s contact mechanics or that of Barber’s.

The results presented do not include the variations of subsurface stresses in the y direction. Please include those as well. Can also author show the contour maps at a cross section below the surface against x and y coordinates?

The results are shown for sigma_xx, sigma xz, and sigma xz. Please also show the results for the other stress components.

In equation (24) is it assumed that E1=2G1 and E2=2G2?, Also isn’t that the Poisson’s ratios should be raised to the power of two?

Can the author provide some validation results?

It would be interesting to see the effect of assumed coefficient of friction value on the maximum developed subsurface stress in terms of for example von Mises stress and also the location of the maximum stress beneath the surface. Would be great if you could please provide this explicitly in the text or add as a table. Another interesting addition would be to see how much they change with the change in the coefficient of friction, if for instance mu was altered from 0.1 to 0.5. Would the relationships be linear or nonlinear and if the latter then to what extent?

The author needs to discuss the limitation of the current solution in the conclusion section and provide information about what is the next step as the continuation of this research in order to get more comprehensive analytical solutions.

Comments on the Quality of English Language

It is acceptable.

Author Response

Please see the attached answer file.

Reviewer 2 Report

Comments and Suggestions for Authors

Elastic stress field beneath a sticking circular contact under tangential load.

                               E. Wellert

The present paper is devoted to the elastic stress analysis in a semi-space domain loaded by a rigid punch inducing a circular contact zone, also axisymmetric stress and displacement states. For punch sticking to the substrate

and executing normal and tangential displacement the contact pressure distribution is related to its contact form z= f(r) defining the contact gap function. In the paper only  the flat punch form is assumed inducing singular  contact pressure distribution. The analytical expressions of stress distribution in the substrate are derived by applying the axisymmetric harmonic potential function taken from Ref. [21]. The closed form presentation is convenient for the analysis, and also the displacement field should be analytically presented.

The second example is referred to the incremental procedure for the rigid punch action with account for the frictional contact response in the presence of sticking and slip zones.

1.The title of paper should be changed to: “Elastic stress field beneath a sticking circular contact under normal and tangential loads”.

2.The analysis presented in Sect. 3 and 4 is related to a problem of very narrow scope, limited only to a rigid, flat ended punch. The gap function f(r) should be introduced for punch shape forms and with both regular and singular stress regimes expressed analytically. The subject matter presentation lacks clarity . It should be explained that the displacement boundary conditions at the contact are determined by the punch contact form. The analytical solutions for  stress distribution should be presented for a wide class of contact shapes.

3. Section 5  first refers to two elastic bodies with contact interaction under normal and tangential loads with no coupling between these two modes, Eq. (22). Next some formulae are taken from Ref. [17] with no explanation. The substrate stress is expressed  by the integral along the loading history, Eq.(29) and (32). Finally, the solution is presented for a rigid flat punch with rounded corners, shown in Fig.5 but no details of solution are shown. It is hard to follow the analysis presented. For the rigid punch the slip zone cannot be activated. There is no logical and consistent development of the analyzed subject.

The paper cannot be recommended for publication in the present form. It should be rewritten and  resubmitted.

Comments on the Quality of English Language

The English text needs corrections in some places

Author Response

Please see the attached answer file.

Round 2

Reviewer 1 Report

Comments and Suggestions for Authors

Thank you for implementing my comments. A very good piece of work!

Author Response

I am very grateful for the kind words of the reviewer.

No suggestions were made to further improve the manuscript.

Reviewer 2 Report

Comments and Suggestions for Authors

The paper in the present form has been slightly improved; but regrettably the review requests have been neglected.

1. The title of paper should be: "Elastic stress field beneath  circular contact under oblique load". Three loading histories should be discussed. First , application of the normal load and next increasing tangential load at constant normal load, second, application of inclined load increasing  at constant inclination angle. Third, frictional sliding contact under normal load and induced tangential displacement. 

2. The stress distribution formulae, such as Eqs. 6, 8 and others are expressed in terms of complex-valued fields. The transition to real variables r, z, or x, y requires further work. Similar remark pertains to the displacement fields. The stress and displacement fields should be expressed in terms of real variables

Comments on the Quality of English Language

Minor editing is needed

Author Response

Comment 2.1: The title of paper should be: "Elastic stress field beneath circular contact under oblique load". Three loading histories should be discussed. First, application of the normal load and next increasing tangential load at constant normal load, second, application of inclined load increasing at constant inclination angle. Third, frictional sliding contact under normal load and induced tangential displacement.

Answer: The core of the manuscript are the exact solutions for the fields in Sects. 3 and 4, especially for the tangential load. The construction of arbitrary axisymmetric frictional contacts of (elastically similar, convex) elastic bodies from the superposition of rigid translations of circular contact domains is a classical procedure that is only reiterated briefly in Sect. 5 to demonstrate, how the core solutions of Sects. 3 and 4 can be used for a broad class of contact mechanical problems.

The discussion of different loading histories (in Sect. 5) would make the manuscript unnecessarily and excessively long, in my opinion. Also, the corresponding contact solutions are known (and best formulated in terms of the method of dimensionality reduction, MDR), and it is clearly stated in Sect. 5, how, for more general loading histories, the subsurface stress field can be obtained from these known contact solutions (in terms of the MDR). Therefore, executing this step is an elementary task, which, in my opinion, would even hurt the understandability and coherence of the core contents of the manuscript.

Comment 2.2: The stress distribution formulae, such as Eqs. 6, 8 and others are expressed in terms of complex-valued fields. The transition to real variables r, z, or x, y requires further work. Similar remark pertains to the displacement fields. The stress and displacement fields should be expressed in terms of real variables.

Answer: Building the real or imaginary part of a complex number is a standard operation, which is a built-in symbolic or numerical function in any advanced computational math environment, like Mathematica or Matlab. Therefore, I consider the given solutions in terms of complex-valued fields as finished and in closed form, although it would probably be possible to rewrite the expressions in terms of real-valued functions of real variables. However, the resulting expressions would be very lengthy and unclear; so, I don’t see the benefit of actually writing them out explicitly.

For the stress field in the surface, it has only been done, because the resulting expressions are still quite compact, and to easily compare them to the corresponding known results.